## [Transparent Peer Review file · Nature Communications]

Predator-mediated local convergence fosters global microbial community divergence

Corresponding Author: Dr Rasit Asiloglu

Version 0:

Reviewer comments:

Reviewer #1

(Remarks to the Author)

The manuscript by Asiloglu et al. investigates the effects of bacterivorous protists (predators) on bacterial community assembly. Through large-scale correlation analyses, microcosm experiments, and synthetic community experiments, the authors aim to demonstrate that predator identity and prey susceptibility jointly determine convergence outcomes. While the topic is interesting and relevant to microbial ecology, the manuscript suffers from several critical weaknesses that undermine the validity of its conclusions.

Overall, this study attempts to infer causal relationships between predator activity and bacterial community assembly from global meta-analyses based primarily on correlation data. However, although predators can indeed regulate bacterial communities, they are also influenced by bacterial composition. Thus, describing predators as drivers of bacterial assembly appears overstated; terms such as predator-mediated assembly or predator–bacteria interactions would be more appropriate.

The authors use the relative abundances of ubiquitous and endemic predators to represent “species-specific” predation, but the rationale for this assumption requires detailed justification. Predator abundance and distribution are shaped by both environmental factors and stochastic processes, and there is currently no direct evidence that endemic predators are truly species-specific. Moreover, the authors do not specify which bacterial taxa correspond to these so-called species-specific predators.

Using a random forest model, the authors report that the abundance of ubiquitous predators, along with pH, aridity, and temperature, predicts bacterial community divergence. However, the presented results (Extended Data Fig. 2) clearly show strong correlations among these environmental variables and predator abundance. Although the authors conducted controlled experiments showing that predator addition alters community richness and dominant taxa, they did not indicate whether the three introduced predators belong to the ubiquitous or endemic groups, preventing meaningful linkage with the earlier findings. Furthermore, the study lacks post-experimental analyses to confirm whether the introduced predators successfully colonized.

Regarding manuscript organization, the Results section is difficult to follow. The research design is not clearly described, particularly in the experimental validation part. Readers must frequently refer to the Methods section to understand the analytical logic, which hinders readability. The authors should explicitly describe the analytical approach and experimental design within each subsection of the Results.

Specific comments

Line 81: The authors state that environmental factors have opposite effects on endemic and ubiquitous predators. This is expected, as by definition the abundances of these two groups are negatively correlated.

Line 84: The term “impact” is inappropriate here, as the analysis only demonstrates correlations, not causality.

Line 89: Correlation alone does not demonstrate that predators influence bacterial community assembly.

Line 93: The correlation between predator abundance and community divergence does not imply that predators drive convergence by suppressing dominant taxa.

Lines 97–99: Although divergence is sensitive to richness, both richness and taxonomic unevenness contribute to divergence. The authors should explain how the results in Fig. 1g–h support the inference that “predation disrupts dominance by a few taxa.” Moreover, the correlation analysis cannot demonstrate that predation enhances bacterial richness.

Lines 132–135: The authors introduced community divergence through fertilizer and CaCO₃ (pH adjustment) treatments and observed similar correlation patterns. They infer that predators tend to converge bacterial communities or at least reduce nutrient- and pH-driven divergence, while endemic predators have no divergence impact due to their low abundance.

However, these experiments cannot establish causality—nutrients and pH may also regulate predator distributions indirectly through their effects on bacterial communities.

Line 151: “a total of 2,121 ASVs and an average richness of 88.9 ± 25.1 ” does not necessarily indicate that all communities are “diverse” or “rich.” The authors should define the thresholds for these terms.

Line 169: Although differences exist between the Hg, Ac, and Av groups, it is not reliable to conclude that taxonomy-based, species-specific effects lead to global divergence based solely on these data.

Line 173: The results in Fig. 3j and Fig. 3h are redundant; as the number of dominant species increases, the relative abundance of each naturally decreases.

Line 189: In the synthetic community experiments, the authors selected predator-resistant and predator-vulnerable bacterial species based on the microcosm results. However, co-culture experiments should be included to verify whether these strains are indeed resistant or vulnerable to the tested predators.

Reviewer #2

(Remarks to the Author)

This is an interesting article aiming to characterise the effects of protist predators on bacterial communities. This is an under researched topic and so any new information here is welcome. Though this MS is somewhat focussed on soil communities, predators are highly likely to have important impacts on microbiomes across habitats, and so publication in a broad reach journal such as *Ncomms* is appropriate - ie the approaches to evaluate predation impacts and the novel findings are likely to be more broadly of interest.

The main strength of the manuscript is that it uses observational omic data from distributed and local soils to ascertain associative effects supporting conclusions; but then goes further to experimentally test mechanisms in manipulated communities. Convincing ecologically grounded arguments are used to explain patterns, which as stated may be broadly applicable. The finding that predators can influence microbiome composition is not in itself novel, though more evidence here is welcomed. Where this manuscript goes further is to suggest more widespread differences in predator impact, and a mechanism explaining context specificity - namely that communities dominated by predator resistant taxa are less likely to be impacted by predation pressure.

My main concern is regarding the early correlational approaches, though this is mitigated by the later experimental work. Firstly, based on an unpublished observation of my own - it is apparent that any molecular community metric derived from an environmental DNA sample will be as good as or better predictor of variance in another molecular metric, compared with co located environmental data. This could be using fungal, plant or protist amplicon data to predict bacterial amplicons. For soils the main environmental predictor is pH; which affects diversity, dominance, composition and variance. The invoking of specific types of interaction between bacteria and protist here is something of a stretch, as the communities could just be co-varying with environment. The authors themselves note in the MS that protists are also structured strongly by soil properties, so I wonder whether the language should be more cautious in the early part of the results.

For example: Lines 140-143 relating to SEM.

Bacterial community change relates to bacterial community dominance: This is known and most likely due to pH effects - composition and diversity metrics tend to relate.

Then, bacterial dominance in the SEM is driven by relative abundances of ubiquitous predators, and community composition of predators. The authors state this provides "strong support for the convergence hypothesis" but what if these community attributes just co vary, eg acidity selects for dominance of both predators and prey? The authors do state that this is not causal, but I think "strong support" is misleading and it would be useful to change to "some" and provide some more text specifically indicating other potential reasons for the observations. Indeed I feel this would only strengthen the case for the experimental work.

It is also possible that a "null" hypothesis could be trialled adding more support for the earlier results - ie picking lineages of 18S amplicons not known to predate bacteria and ascertaining their contributions to bacterial change - "divergence" or "convergence". I suspect strong relationships either way can be found for many non-predatory taxa, or do protists uniquely explain more variance in bacterial communities?

The next experimental results are I think fantastic - figures 3e-3g are marvellous. Really clear demonstration of both the impact of protists on community development as well as signifying differing effects of predators on specific taxa, and how this translates to community level change. Similarly the syncomm experiment goes further to support the manuscripts overall claims.

Whilst the manuscript is generally well written and presented some careful checking of English may be required at proofing.

Reviewer #3

(Remarks to the Author)

This excellent manuscript deals with the effect of predator protists on bacterial communities diversity. They test two scenarios, if predation tends to reduce or increase their diversity. In that purpose, they first use global meta-analyses, local

field experimentation, microcosms and in vitro experiments with known strains. Results show that predators tend to reduce diversity locally by preying on the most common species, but promote diversity at large scale, due to local specialization and trait-based interactions. These results show that bacterial communities (and, therefore, functions) can be steered in a predictable way through the use of a certain protist species. This finding opens the way for soil microbiome engineering, which would then have many practical applications like disease suppression or soil fertility increase. For these reasons, I recommend the publication of this article.

The article is well written and easy to follow. I still have a few comments though

Line 74: Here I would specify the marker gene (probably 18S v9)

Line 77: The gene coding for the 18S rRNA does not allow separating between species, and especially the v9 region. Of course, one has to take the available data but one has to bear in mind that each 18S ASV represents rather a group of related species that can potentially have the same ecology, food regime, etc... or not! This is one of the potential limitations of the study, which can be discussed.

Likewise, "endemic" is generally used for organisms that are restricted to a certain geographical area. The word "ecosystem" does not apply to, for instance, temperate grasslands; these would be "ecosystem types" or biomes.

Line 78: Because the Methods section is at the end of the manuscript, it is difficult to know how the RA have been calculated. I would advise to drop one line about it, or simply to refer directly to the Materials section.

Line 116: here one could add "by increasing the number of available substrates"

Line 117: Are the ubiquitous protist ASVs the same ones as in the former experiment, or have they been reclassified?

Line 129: between (instead of "of")

Line 169: I am not convinced that taxonomy has a lot to do here. *Acanthamoeba* (Discosea) is very distantly related to *Vermamoeba* (Tubulinea). I would suggest that both taxa feed on bacteria using their pseudopodia, while "*Heteromita*" uses its flagella, and therefore captures bacteria that are characterized by other traits. The difference would be more functional than taxonomic.

Fig 5: "Vulnerable"; what is the difference between a and b?

Line 474: UniFrac distances include a phylogenetic component, and given that the manuscript is oriented towards calculating dissimilarities, these distances would be interesting to apply as well. Maybe the results do not differ substantially, but I think that giving it a preliminary try (and maybe more if it's worth doing it) would be a good idea. UniFrac does also have a quantitative component.

Lines 541-543: It would be useful to know, at least roughly, which taxa are retained as predator and non-predator protists for the calculation of the RA. Or is it the ratio of heterotrophic protists versus all eukaryotes (including fungi animals and plants)? How did the authors determine that their ASVs correspond to organisms that prey on protists; did they rely on a functional database?

Line 571: This taxon name "*Heteromita*" used to include an immense diversity of taxa (basically almost all Glissomonadida) has been invalidated. I guess that reference sequences for these three cultures have been published elsewhere, but I would provide the at least their GenBank accession codes, and name that species according to its closest relative. Otherwise, this would prevent reproducibility of the experiments.

Line 618: *Knoellia sinensis* (with "s"); by the way, the fast grower Pt is capable of forming spores; how much would that influence the experience?

Version 1:

Reviewer comments:

Reviewer #1

(Remarks to the Author)

I appreciate the authors' careful revisions and detailed responses to my earlier comments. The manuscript has been substantially improved, and the major concerns regarding the interpretation of predator-bacteria interactions raised during my initial review have been satisfactorily addressed.

I recommend acceptance after the authors address the following minor formatting issue.

In Fig. 3C, the font used for the x-axis label and the genus names appears non-standard, and some letters overlap when viewed on my system. Please adjust the font style to ensure these labels render properly, and italicize the genus names. After this minor formatting adjustment, the manuscript does not need to be returned to me for further review.

Reviewer #2

(Remarks to the Author)

I am happy with the authors amendments, have no further comments and recommend the article suitable for publication.

Reviewer #3

(Remarks to the Author)

I have read carefully all the answers provided, and I think that most comments have been reasonably addressed. There is only one (small) point on which I disagree, is to keep the name "*Heteromita*". This species name has been invalidated; I would instead name it "soil glissomonad", pending a formal description of the group.

Besides this very small detail, I think that the manuscript is ready for publication.

RESPONSE TO THE REVIEWER COMMENTS

Reviewer #1 (Remarks to the Author):

The manuscript by Asiloglu et al. investigates the effects of bacterivorous protists (predators) on bacterial community assembly. Through large-scale correlation analyses, microcosm experiments, and synthetic community experiments, the authors aim to demonstrate that predator identity and prey susceptibility jointly determine convergence outcomes. While the topic is interesting and relevant to microbial ecology, the manuscript suffers from several critical weaknesses that undermine the validity of its conclusions.

Answer: Thank you for taking the time to evaluate our manuscript. We appreciate your constructive criticism. We revised the manuscript according to your and other reviewers' comments, as shown below (Blue text represents our responses, and the line numbers of the final manuscript file is provided to show the changes). Briefly, we replaced overstated causal language on correlation-based results (e.g., "predator-driven") with more appropriate terms such as "predator-mediated", clarified the rationale and classification of predator types, and incorporated additional explanations of experimental design directly within the Results section. We also added methodological details, addressed limitations of sequence resolution, and refined the English throughout the text to enhance readability and transparency. In addition, we adhered to the journal's formatting requirements. Briefly, the abstract was shortened to 150 words, subheadings were organized, and references were adjusted.

Overall, this study attempts to infer causal relationships between predator activity and bacterial community assembly from global meta-analyses based primarily on correlation data. However, although predators can indeed regulate bacterial communities, they are also influenced by bacterial composition. Thus, describing predators as drivers of bacterial assembly appears overstated; terms such as predator-mediated assembly or predator–bacteria interactions would be more appropriate.

Answer: Thank you for the comment. We have reviewed and revised any overstated descriptions of predator effects. The term "predator-driven" has been replaced with "predator-mediated" throughout the manuscript, including in the title. We have removed causative language where results are correlational. All revisions have been highlighted in blue in the revised version of the manuscript for clarity.

The authors use the relative abundances of ubiquitous and endemic predators to represent "species-specific" predation, but the rationale for this assumption requires detailed justification. Predator abundance and distribution are shaped by both environmental factors and stochastic processes, and there is currently no direct evidence that endemic predators are truly species-specific. Moreover, the authors do not specify which bacterial taxa correspond to these so-called species-specific predators. Using a random forest model, the authors report that the abundance of ubiquitous predators, along with pH, aridity, and temperature, predicts bacterial community divergence. However, the presented results (Supplementary Fig. 2) clearly show strong correlations among these environmental variables and predator abundance.

Answer: Thank you for the comment. We agree that environmental factors may similarly influence both bacterial and protist communities. We used correlation-based results to disentangle the relationship between predator abundance and bacterial community divergence, which was later confirmed in microcosm studies. We had a discussion on this matter in the submitted initially manuscript under the subsection "Net convergence effect of predators". However, to enhance clarity, the following discussion was added to the main text.

L153-157: "Indeed, the effects of environmental factors such as pH and nutrient levels on predator diversity and abundances (Supplementary Figs 2 and 5) cannot be excluded in both global and local field studies. To establish causality and isolate predator effects from environmental drivers, we subsequently conducted a complementary *in vitro* microcosm experiment under defined conditions with and without predator introductions."

Although the authors conducted controlled experiments showing that predator addition alters community richness and dominant taxa, they did not indicate whether the three introduced predators belong to the ubiquitous or endemic groups, preventing meaningful linkage with the earlier findings.

Answer: Thank you for this helpful comment. The partial amplification of the 18S rRNA gene in our study does not allow precise taxonomic-level identification of environmental protists at the strain level. However, the introduced predators — *Acanthamoeba castellanii*, *Vermamoeba vermiformis*, and *Heteromita globosa* — are well-documented common soil protists reported from a wide range of terrestrial ecosystems. Therefore, we consider them to represent ubiquitous predator taxa rather than endemic ones.

The following sentence was added to the main text for clarification:

L157-159: “The introduced predators, *Acanthamoeba castellanii* (Ac), *Heteromita globosa* (Hg), and *Vermamoeba vermiformis* (Vv), are widely distributed soil protists commonly found across diverse terrestrial habitats, and thus considered as ubiquitous organisms.”

Furthermore, the study lacks post-experimental analyses to confirm whether the introduced predators successfully colonized.

Answer: Although we do not have exact numbers of protists at the end of the experiment, we did check the successful colonization of the introduced protists. The methodology was added to the manuscript.

L408-410: “1 g of calcined clay was transferred to a sterile tube. The 1000 × dilution of the sample were inspected with an inverted microscope at ×200 and ×400 magnifications (Nikon Eclipse TE2000-S, Tokyo, Japan) to check the successful colonisation of the protists.”

Regarding manuscript organization, the Results section is difficult to follow. The research design is not clearly described, particularly in the experimental validation part. Readers must frequently refer to the Methods section to understand the analytical logic, which hinders readability. The authors should explicitly describe the analytical approach and experimental design within each subsection of the Results.

Answer: Thank you for the critical comment. To enhance the readability, we added explanations to each subsection of the Results.

L72-74: “The soils included in this study represent a wide range of vegetation types (shrublands, grassland, and forest), edaphic characteristics such as pH, water and nutrient contents, and bioclimatic regions (arid, temperate, tropical, and continental).”

L121-124: “All treatments received chemical fertiliser, while organic amendments such as filter cake, cattle manure, bagasse, and wood biochar were applied at different rates (5 or 10 tons C ha⁻¹) to manipulate soil resource availability (Supplementary Tables 1–2). Soil samples were taken from each field at 0–10 cm depth and each treatment was represented with four replicates.”

L157-166: “The introduced predators, *Acanthamoeba castellanii* (Ac), *Heteromita globosa* (Hg), and *Vermamoeba vermiformis* (Vv), are widely distributed soil protists commonly found across diverse terrestrial habitats, and thus considered as ubiquitous organisms. We obtained indigenous predator-free bacterial communities from five distinct soil types: forest (Com1), grassland (Com2), paddy field (Com3), soybean field (Com4), and sugarcane field (Com5). The experimental design consisted of the four protist treatments (Control with no protist addition, Ac, Hg, and Vv) for each of the five bacterial communities, making a total of 20 microcosms with six replicates each. The microorganisms were incubated in calcined clay, an inert soil substitute^{30,31}, under defined nutrient conditions for 5

weeks. This design enabled us to directly assess the exclusive role of individual predator species in driving bacterial convergence, independent of environmental heterogeneity²¹. We descriptively sampled three microcosms on the 3rd and 5th weeks.”

L205-211: The first SynCom received equal abundances of all six bacterial species (1.8×10^6 cells g calcined clay⁻¹), while the other six each had one dominant species (1.0×10^7 cells g calcined clay⁻¹) and five rare ones (1.0×10^5 cells g calcined clay⁻¹). All SynComs had the same total amount of bacteria (1.1×10^7 cells g calcined clay⁻¹) (Fig. 4a). The experimental design consisted of the four protist treatments (Ctrl, Ac, Hg, and Vv) for each of the seven synthetic bacterial communities, making a total of 28 microcosms with three replicates each. The microorganisms were incubated in calcined clay, an inert soil substitute^{30,31}, under defined nutrient conditions. We descriptively sampled the microcosms in the 3rd week.

Specific comments

Line 81: The authors state that environmental factors have opposite effects on endemic and ubiquitous predators. This is expected, as by definition the abundances of these two groups are negatively correlated.

Answer: Thank you for the comment. We agree that, by definition, the abundances of ubiquitous and endemic groups are expected to be negatively correlated. However, in our study, we defined ubiquitous ASVs as those detected in more than 60% of samples and endemic ASVs as those detected in fewer than 20% of samples (as described in Lines 81-82 for clarity). Therefore, these two groups do not necessarily show opposing patterns, since approximately 20% of ASVs fall between these thresholds and are not included in either category. Consequently, the relative abundances of ubiquitous and endemic predators are not expected to sum to 100%. Indeed, in the second experiment (the local field trial), these groups did not exhibit opposing responses to environmental factors (Supplementary Fig. S5), supporting this interpretation.

Line 84: The term “impact” is inappropriate here, as the analysis only demonstrates correlations, not causality.

Answer: Thank you. We have softened our interpretation as follows.

L86-87: “indicating the possibility of their effect on community convergence”

Line 89: Correlation alone does not demonstrate that predators influence bacterial community assembly.

Answer: Thank you for the comment. We agree that correlation alone does not demonstrate causality. In our study, this correlation-based result is intended to show an association rather than a direct causal relationship. To address this point, we have revised the sentence. The following modification clarifies that our interpretation is correlative and not a demonstration of direct causation.

L92-95: “These results may suggest a dual role of predators: while the global divergence of bacterial communities may arise from differences in the composition of predator communities across ecosystems through likely their species-specific effects, ubiquitous predators may mediate bacterial community converge by exerting consistent effects across the globe.”

Line 93: The correlation between predator abundance and community divergence does not imply that predators drive convergence by suppressing dominant taxa.

Answer: Thank you for the comment. We agree that the correlation between predator abundance and community divergence does not, in itself, demonstrate that predators drive convergence by suppressing dominant taxa. However, the observed correlation supports “the convergence hypothesis (Fig. 1a)”, which proposes that convergence may occur due to predators suppressing dominant taxa. Therefore, we further examined the relationship between the richness of dominant bacterial taxa and the relative abundance of ubiquitous predators to explore this potential mechanism. At this stage of the manuscript, we do not interpret the correlation as direct evidence of causality but rather as a basis for hypothesis-driven exploration. We revised the sentence for clarification.

L96-98: “Although correlation-based, the initial results support the convergence hypothesis—that is, convergence may occur when predators suppress dominant taxa—rather than the divergence hypothesis. Therefore, we further examined the relationship between the richness of dominant bacterial taxa and the relative abundance of ubiquitous predators.”

Lines 97–99: Although divergence is sensitive to richness, both richness and taxonomic unevenness contribute to divergence. The authors should explain how the results in Fig. 1g–h support the inference that “predation disrupts dominance by a few taxa.” Moreover, the correlation analysis cannot demonstrate that predation enhances bacterial richness.

Answer: Thank you for the comment. We agree that both richness and taxonomic unevenness contribute to community divergence and that correlation alone cannot demonstrate causality. In Fig. 1g–h, we observed that the relative abundance of ubiquitous predators was positively correlated with the richness of dominant bacterial taxa. Here, “dominant” taxa refer to bacterial ASVs with relative abundances exceeding 1%. This pattern suggests that in communities with higher predator abundance, more dominant bacterial taxa coexist, implying that predation may alleviate competitive exclusion among a few highly abundant species. In other words, predation likely reduces the disproportionate dominance of a small subset of bacteria, allowing more taxa to persist above the dominance threshold. This mechanism aligns with the inference that “predation disrupts dominance by a few taxa.” However, we acknowledge that our statement that “predation disrupts dominance by a few taxa” was based on the observed positive relationship and this relationship does not establish a direct causal link. To clarify this interpretation, we have revised the relevant sentence in the main text as follows:

L101-103: “We found that ubiquitous predators were positively associated with the richness of the dominant bacterial taxa (Fig. 1g–h), suggesting that predation may alleviate dominance by a few taxa and potentially promote dominance by broader range of species.”

Lines 132–135: The authors introduced community divergence through fertiliser and CaCO₃ (pH adjustment) treatments and observed similar correlation patterns. They infer that predators tend to converge bacterial communities or at least reduce nutrient- and pH-driven divergence, while endemic predators have no divergence impact due to their low abundance. However, these experiments cannot establish causality—nutrients and pH may also regulate predator distributions indirectly through their effects on bacterial communities.

Answer: Thank you for the constructive comment. We agree that our correlation-based results cannot establish causality and that nutrient and pH treatments may also influence predator distributions. Nevertheless, the consistent correlation patterns observed across different environmental manipulations make the convergence hypothesis (Fig. 1a) more plausible as an explanation for our results. To clarify this point, we have revised the sentence to emphasize that our findings are consistent with, but do not prove, this hypothesis.

L141-144: “Taken together, these correlation-based results make the convergence hypothesis more plausible, suggesting that under local conditions where predator taxa are similar and abundant, they may contribute to bacterial community convergence or reduce nutrient- and pH-driven divergence, although nutrient and pH effects on predators cannot be excluded.”

We have also added text in the following section acknowledging the potential effects of environmental factors and explaining that causality was further tested through a controlled microcosm experiment.

L151-157: “Although these results provided that prey-predator interaction potentially leading to the convergence hypothesis, they did not offer definitive causal evidence due to the background of environmental variability. Indeed, the effects of environmental factors, such as pH and nutrient levels, on predator diversity and abundance (Supplementary Figs 2 and 5) cannot be excluded in both global and local field studies. To establish causality and isolate predatory effects from environmental drivers, we subsequently conducted a complementary *in vitro* microcosm experiment under defined conditions with and without predator introductions.”

Line 151: “a total of 2,121 ASVs and an average richness of 88.9 ± 25.1 ” does not necessarily indicate that all communities are “diverse” or “rich.” The authors should define the thresholds for these terms.

Answer: We have revised the sentence as follows.

L167-168: “Bacterial communities consisted of a total of 2121 ASVs with an average richness of 88.9 ± 25.1 , indicating successful colonisation of the native communities.”

Line 169: Although differences exist between the Hg, Ac, and Av groups, it is not reliable to conclude that taxonomy-based, species-specific effects lead to global divergence based solely on these data.

Answer: Thank you. Indeed, traits such as the size and feeding type of the introduced predators differ. We agree with you. We deleted the speculative section, and the sentence has been edited as follows.

L184-185: “This may support that local convergence effect of predators can ultimately lead to global divergence, consistent with our expectations and previous findings”

Line 173: The results in Fig. 3j and Fig. 3h are redundant; as the number of dominant species increases, the relative abundance of each naturally decreases.

Answer: Thank you for the comment. We believe there may be a slight misunderstanding regarding Figures 3h and 3j. Although both panels present data related to dominant ASVs, they address different aspects of the results. Figure 3h shows the overall composition and relative abundance of dominant ASVs across all treatments, whereas Figure 3j illustrates explicitly the relative abundance changes of the dominant ASVs identified in the Control treatment. This panel is important for demonstrating that the relative abundance of these dominant Control ASVs decreases under protist treatments, highlighting the suppressive effects of predators on initially dominant taxa.

We would also like to note that Reviewer 2 found these figures (Figs. 3e–3g) to be particularly clear and effective in illustrating the impact of protists on community development and taxa-specific effects. Considering both perspectives, we believe Figures 3h and 3j together provide a comprehensive understanding of how protists shape community structure, and therefore, with all due respect, we prefer to retain the current presentation.

Line 189: In the synthetic community experiments, the authors selected predator-resistant and predator-vulnerable bacterial species based on the microcosm results. However, co-culture experiments should be included to verify whether these strains are indeed resistant or vulnerable to the tested predators.

Answer: Thank you for this insightful comment. Actually, before conducting the synthetic community (SynCom) experiments, we performed preliminary co-culture (pairwise interaction) experiments between each protist species and individual bacterial isolates. These pilot tests were used to assess predation effects and thereby classify the bacterial strains as predator-resistant or predator-vulnerable. The SynCom experiment was subsequently designed after these verified interactions. To clarify this procedure, we have added the following sentence to the Methods section:

L449-451: "Prior to the SynCom experiment, we conducted pilot co-culture assays between each protist and bacterial isolate to confirm their resistance or vulnerability to predation, and these results were used to confirm predator-resistant and predator-vulnerable species."

Reviewer #2 (Remarks to the Author):

This is an interesting article aiming to characterise the effects of protist predators on bacterial communities. This is an under researched topic and so any new information here is welcome. Though this MS is somewhat focussed on soil communities, predators are highly likely to have important impacts on microbiomes across habitats, and so publication in a broad reach journal such as *Ncomms* is appropriate - ie the approaches to evaluate predation impacts and the novel findings are likely to be more broadly of interest.

The main strength of the manuscript is that it uses observational omic data from distributed and local soils to ascertain associative effects supporting conclusions; but then goes further to experimentally test mechanisms in manipulated communities. Convincing ecologically grounded arguments are used to explain patterns, which as stated may be broadly applicable. The finding that predators can influence microbiome composition is not in itself novel, though more evidence here is welcomed. Where this manuscript goes further is to suggest more widespread differences in predator impact, and a mechanism explaining context specificity - namely that communities dominated by predator resistant taxa are less likely to be impacted by predation pressure.

Answer: Thank you for taking your time to evaluate our manuscript. We appreciate your constructive criticism. We revised the manuscript according to your and other reviewers' comments, as shown below (Blue text represents our responses, and the line numbers of the final manuscript file is provided to show the changes). Briefly, we replaced overstated causal language on correlation-based results (e.g., "predator-driven") with more appropriate terms such as "predator-mediated," clarified the rationale and classification of predator types, and incorporated additional explanations of experimental design directly within the Results section. We also added methodological details, addressed limitations of sequence resolution, and refined the English throughout the text to enhance readability and transparency. In addition, we adhered the manuscript to the journal's formatting requirements. Briefly, the abstract was shortened to 150 words, subheadings were organized, and references were adjusted.

My main concern is regarding the early correlational approaches, though this is mitigated by the later experimental work. Firstly, based on an unpublished observation of my own - it is apparent that any molecular community metric derived from an environmental DNA sample will be as good as or better predictor of variance in another molecular metric, compared with co located environmental data. This could be using fungal, plant or protist amplicon data to predict bacterial amplicons. For soils the main environmental predictor is pH; which affects diversity, dominance, composition and variance. The invoking of specific types of interaction between bacteria and protist here is something of a stretch, as the communities could just be co-varying with environment. The authors themselves note in the MS that protists are also structured strongly by soil properties, so I wonder whether the language should be more cautious in the early part of the results. For example: Lines 140-143 relating to SEM. Bacterial community change relates to bacterial community dominance: This is known and most likely due to pH effects - composition and diversity metrics tend to relate. Then, bacterial dominance in the SEM is driven by relative abundances of ubiquitous predators, and

community composition of predators. The authors state this provides "strong support for the convergence hypothesis" but what if these community attributes just co vary, eg acidity selects for dominance of both predators and prey? The authors do state that this is not causal, but I think "strong support" is misleading and it would be useful to change to "some" and provide some more text specifically indicating other potential reasons for the observations. Indeed I feel this would only strengthen the case for the experimental work.

Answer: Thank you for the insightful comment. We agree with you on that protist and bacterial data deriving from the same DNA may overestimate the correlation-based results. We obtained similar comments from other reviewers as well. We have reviewed and revised any overstated descriptions of predator effects. All revisions have been highlighted in blue in the revised version of the manuscript for clarity. Briefly, the term "predator-driven" has been replaced with "predator-mediated" throughout the manuscript, including in the title. We agree that correlation does not prove causality; therefore, we have removed causative language when results are correlational throughout the manuscript. Discussion was added regarding the potential effects of environmental factors on predators.

L141-144: "Taken together, these correlation-based results make the convergence hypothesis more plausible, suggesting that under local conditions where predator taxa are similar and abundant, they may contribute to bacterial community convergence or reduce nutrient- and pH-driven divergence, although nutrient and pH effects on predators cannot be excluded."

We have also added text in the following section acknowledging the potential effects of environmental factors and explaining that causality was further tested through a controlled microcosm experiment.

L151-157: "Although these results provided that prey-predator interaction potentially leading to the convergence hypothesis, they did not offer definitive causal evidence due to the background of environmental variability. Indeed, the effects of environmental factors, such as pH and nutrient levels, on predator diversity and abundance (Supplementary Figs 2 and 5) cannot be excluded in both global and local field studies. To establish causality and isolate predatory effects from environmental drivers, we subsequently conducted a complementary *in vitro* microcosm experiment under defined conditions with and without predator introductions."

It is also possible that a "null" hypothesis could be trialled adding more support for the earlier results - ie picking lineages of 18S amplicons not known to predate bacteria and ascertaining their contributions to bacterial change - "divergence" or "convergence". I suspect strong relationships either way can be found for many non-predatory taxa, or do protists uniquely explain more variance in bacterial communities?

Answer: Thank you for this insightful suggestion. We agree that including a "null" comparison using non-predatory protist lineages would provide an important test to evaluate whether the observed relationships are unique to predatory taxa. However, in our current dataset, the relative abundances of predatory taxa were used as explanatory variables in Studies 1 and 2. Because non-predatory taxa can only be represented as the complementary fraction (i.e., 1 – predatory taxa), testing their relationships with bacterial divergence or convergence would produce statistically redundant results. Therefore, this approach (although we fully agree) cannot effectively distinguish the specific effects of predatory versus non-predatory protists with the available data in this manuscript.

Nonetheless, our subsequent controlled experiments (Studies 3 and 4) were designed to address causality and isolate predator-specific effects under defined conditions. These experiments demonstrated consistent and mechanistically interpretable patterns that strongly support the conclusions derived from the correlative analyses.

We agree that future studies incorporating community-wide functional assignments, such as non-predatory mixotrophs or saprotrophs as controls, will be valuable for testing whether predatory protists uniquely explain bacterial community variance. We plan to incorporate this comparative framework in our future work.

The next experimental results are I think fantastic - figures 3e-3g are marvellous. Really clear demonstration of both the impact of protists on community development as well as signifying differing effects of predators on specific taxa, and how this translates to community level change. Similarly the syncomm experiment goes further to support the manuscripts overall claims. Whilst the manuscript is generally well written and presented some careful checking of English may be required at proofing.

Answer: Thank you very much for the encouraging comments and positive evaluation of our results. The manuscript has been carefully rechecked, and the English language has been further refined throughout.

Reviewer #3 (Remarks to the Author):

This excellent manuscript deals with the effect of predator protists on bacterial communities diversity. They test two scenarios, if predation tends to reduce or increase their diversity. In that purpose, they first use global meta-analyses, local field experimentation, microcosms and in vitro experiments with known strains. Results show that predators tend to reduce diversity locally by preying on the most common species, but promote diversity at large scale, due to local specialization and trait-based interactions. These results show that bacterial communities (and, therefore, functions) can be steered in a predictable way through the use of a certain protist species. This finding opens the way for soil microbiome engineering, which would then have many practical applications like disease suppression or soil fertility increase. For these reasons, I recommend the publication of this article. The article is well written and easy to follow. I still have a few comments though.

Answer: Thank you very much for the encouraging comments and positive evaluation of our results. We appreciate your constructive criticism. We revised the manuscript according to your and other reviewers' comments, as shown below (Blue text represents our responses, and the line numbers of the final manuscript file is provided to show the changes). Briefly, we replaced overstated causal language on correlation-based results (e.g., "predator-driven") with more appropriate terms such as "predator-mediated," clarified the rationale and classification of predator types, and incorporated additional explanations of experimental design directly within the Results section. We also added methodological details, addressed limitations of sequence resolution, and refined the English throughout the text to enhance readability and transparency. In addition, we adhered the manuscript to journal's formatting requirements. Briefly, the abstract was shortened to 150 words, subheadings were organized, and references were adjusted.

Line 74: Here I would specify the marker gene (probably 18S v9)

Answer: Thank you. Information of the marker genes and primer sets is added for both global and local studies as follows.

L74-75: "Predators were extracted from publicly available amplicon sequence variant (ASV) table, which was generated by amplifying the V4-V5 region of the 18S rRNA gene using the 616*F/1132R primer set (methods, Supplementary Fig. 1a-b)."

L130-131: "Protist data were obtained by amplifying the V9 region of the 18S rRNA gene using the 1389F/1510R primer set."

Line 77: The gene coding for the 18S rRNA does not allow separating between species, and especially the v9 region. Of course, one has to take the available data but one has to bear in mind that each 18S ASV represents rather a group

of related species that can potentially have the same ecology, food regime, etc... or not! This is one of the potential limitations of the study, which can be discussed.

Answer: Thank you for the comment. The V9 region was only used in the local island study. We added the following section to the conclusion to address the potential limitation.

L259-263: “A potential limitation of our study is that the 18S rRNA gene, particularly V9 region used in local field study, may not allow reliable discrimination at the species level. Consequently, each ASV may represent a group of related protist species that could differ in ecology, prey preferences, or functional traits. While this limits the resolution of our predator–prey assignments, our multi-scale experimental approach, including microcosms and synthetic communities, helps mitigate this limitation by directly testing predatory effects.”

Likewise, "endemic" is generally used for organisms that are restricted to a certain geographical area. The word "ecosystem" does not apply to, for instance, temperate grasslands; these would be "ecosystem types" or biomes.

Answer: We have redefined the usage of “endemic” in this manuscript.

L81-82: “endemic (ASVs presented in less than 20% of the samples)”.

Line 78: Because the Methods section is at the end of the manuscript, it is difficult to know how the RA have been calculated. I would advise to drop one line about it, or simply to refer directly to the Materials section.

Answer: Thank you. We added an explanation on calculation of predator RA.

L283-284: “The relative abundance (RA) of predators used in all steps of our study represented their abundance distribution among all protists, which was comparable with the previous study.”

Line 116: here one could add "by increasing the number of available substrates"

Answer: Done (L120-121).

Line 117: Are the ubiquitous protist ASVs the same ones as in the former experiment, or have they been reclassified?

Answer: In each study, we reclassified the ubiquitous protists, which represent ASVs found in more than 60% of the samples.

Line 129: between (instead of “of”)

Answer: Done.

Line 169: I am not convinced that taxonomy has a lot to do here. Acanthamoeba (Discosea) is very distantly related to Vermamoeba (Tubulinea). I would suggest that both taxa feed on bacteria using their pseudopodia, while "Heteromita" uses its flagella, and therefore captures bacteria that are characterized by other traits. The difference would be more functional than taxonomic.

Answer: Thank you. Indeed, traits such as the size and feeding type of the introduced predators differ. We agree with you. We deleted the speculative section, and the sentence has been edited as follows.

L184-185: “This may support that local convergence effect of predators can ultimately lead to global divergence, consistent with our expectations and previous findings.”

Fig 5: “Vulnerable”; what is the difference between a and b?

Answer: We appreciate the comment and would like to clarify the figure.

- If the question refers to the two “vulnerable” bacteria (dark and light red colors): these colors indicate different bacterial species that are vulnerable to predation. Different species may respond differently to the same predator, which is why they are represented separately.
- If the question refers to panels a and b (community A and B), both panels represent protist communities with slightly different compositions; however, the dominant predator is the same in both cases.

Line 474: UniFrac distances include a phylogenetic component, and given that the manuscript is oriented towards calculating dissimilarities, these distances would be interesting to apply as well. Maybe the results do not differ substantially, but I think that giving it a preliminary try (and maybe more if it’s worth doing it) would be a good idea. UniFrac does also have a quantitative component.

Answer: Thank you for this insightful suggestion. We calculated weighted UniFrac distances as recommended and obtained results that were highly consistent with those based on Bray–Curtis dissimilarity. Although minor differences were observed, they did not affect the overall conclusions of the study.

However, as protist feeding preferences are not strictly linked to the taxonomic identity of bacteria — and can vary even among closely related bacterial species — we believe that a taxonomy-independent metric such as Bray–Curtis dissimilarity is more appropriate for capturing predator-driven community changes. Therefore, we would like to present the Bray–Curtis results in the main text.

Lines 541-543: It would be useful to know, at least roughly, which taxa are retained as predator and non-predator protists for the calculation of the RA. Or is it the ratio of heterotrophic protists versus all eukaryotes (including fungi animals and plants)? How did the authors determine that their ASVs correspond to organisms that prey on protists; did they rely on a functional database?

Answer: Thank you for this valuable comment. We manually assigned ASVs to predatory protists based on their taxonomic classification at the genus or family level. This approach requires specific expertise in protist taxonomy and functional ecology, which our group has established through multiple previous studies using the same method (Asiloglu et al., 2021: doi: <https://doi.org/10.1016/j.soilbio.2021.108397>; Bodur et al., 2025, doi: <https://doi.org/10.1016/j.scitotenv.2025.179606>). The original article (Oliverio et al., 2020) from which we obtained the global protist community data also applied this taxon-based functional classification, and our results were comparable with theirs (as mentioned in Line 80).

To clarify this in the manuscript, we have revised the Methods section as follows:

L281-283: “Then functional group of bacterivorous protists (predators) was manually assigned based on their taxonomic annotation following previously established classifications as previously shown^{21,44,45}.”

Line 571: This taxon name “Heteromita” used to include an immense diversity of taxa (basically almost all Glissomonadida) has been invalidated. I guess that reference sequences for these three cultures have been published elsewhere, but I would provide the at least their GenBank accession codes, and name that species according to its closest relative. Otherwise, this would prevent reproducibility of the experiments.

Answer: Thank you for the comment. *Heteromita globosa* LAP3-2017 was isolated by us from a paddy field, and the related information, including GenBank accession codes, was previously published (*) as you mentioned. For clarity, we added the information on the GenBank accession code to the methods section.

L391-392: “The NCBI genbank accession number of *H. globosa* LAP3-2017 is LC764482.”

(*): Fujino, M., Suzuki, K., Harada, N., & Asiloglu, R. (2023). Protists modulate active bacterial community composition in paddy field soils. *Biology and Fertility of Soils*, 59(7), 709-721. doi: <https://doi.org/10.1007/s00374-023-01733-5>

Line 618: *Knoellia sinensis* (with “s”); by the way, the fast grower Pt is capable of forming spores; how much would that influence the experience?

Answer: Thank you. The spelling of *Knoellia sinensis* has been corrected. Sporulation could reduce bacterial susceptibility to predation and contribute to its high abundance in predator treatments. Nonetheless, vulnerable bacterial species consistently decreased, indicating that active predator–prey interactions occurred. Therefore, while sporulation may have partially protected spore-forming bacteria, the overall patterns of predator effects on bacterial community composition remain robust. The following sentence was added to the main text.

L216-218: “We note that some predator-resistant bacteria in our study are capable of sporulation, which may have contributed to their persistence under predation, potentially modulating the degree of convergence observed.”

RESPONSE TO THE REVIEWER COMMENTS

Reviewer #1 (Remarks to the Author):

I appreciate the authors' careful revisions and detailed responses to my earlier comments. The manuscript has been substantially improved, and the major concerns regarding the interpretation of predator-bacteria interactions raised during my initial review have been satisfactorily addressed.

I recommend acceptance after the authors address the following minor formatting issue.

In Fig. 3C, the font used for the x-axis label and the genus names appears non-standard, and some letters overlap when viewed on my system. Please adjust the font style to ensure these labels render properly, and italicize the genus names. After this minor formatting adjustment, the manuscript does not need to be returned to me for further review.

Answer: Thank you very much for taking your time to review our manuscript. We appreciate your efforts and valuable comments. We have adjusted the kerning and italicised the genus names.

Reviewer #2 (Remarks to the Author):

I am happy with the authors ammendments, have no further comments and recommend the article suitable for publication.

Answer: Thank you very much for taking your time to review our manuscript. We appreciate your efforts and valuable comments.

Reviewer #3 (Remarks to the Author):

I have read carefully all the answers provided, and I think that most comments have been reasonably addressed. There is only one (small) point on which I disagree, is to keep the name "Heteromita". This species name has been invalidated; I would instead name it "soil glissomonad", pending a formal description of the group. Besides this very small detail, I think that the manuscript is ready for publication.

Answer: Thank you very much for taking your time to review our manuscript. We appreciate your efforts and valuable comments.

Following your suggestion, we have revised the terminology in the main text as follows.

L158: a soil glissomonad (*Heteromita globosa* [Hg], informal designation pending formal taxonomic description)

L390: a soil glissomonad

L403: Hg, a soil glissomonad (*Heteromita globosa*, informal designation pending formal taxonomic description)